# Virtual World as an Interactive Safety Training Platform

**Sayli Shiradkar** [1,*], **Luis Rabelo** [2], **Fahad Alasim** [3] **and Khalid Nagadi** [4]

1 Industrial Engineering and Operations Research, Indian Institute of Technology Bombay, Mumbai 400076, India
2 Industrial Engineering and Management Systems, University of Central Florida, Orlando, FL 32816, USA; luis.rabelo@ucf.edu
3 Industrial Engineering Department, King Saud University, Riyadh 11421, Saudi Arabia; falasim@ksu.edu.sa
4 Department of Industrial and Systems Engineering, University of Jeddah, Jeddah 23218, Saudi Arabia; knagadi@uj.edu.sa
* Correspondence: sayli.bhide@gmail.com

**Abstract:** Virtual training platform allows interactive and engaging learning through practice without exposing trainees to hazards. In the recent pandemic (COVID-19) situation, online training is gaining importance as it allows learning with social distancing. This research study develops two online training modes—slide-based and virtual world—and assesses them on factors such as knowledge retention, engagement, and attention. Fire safety and emergency evacuation procedures were selected for online training development, focusing on a university community. A Lean Startup methodology was employed to develop training content for virtual and slide-based safety training (SBST). A virtual university building was developed with 15 learning objectives on fire safety. An empirical evaluation of the training modes was conducted with 143 participants. The results validated that a Virtual Safety World (VSW) can provide the same knowledge as SBST but can do so in a more engaging manner. Retention of concepts after a month was higher in VSW participants. The participants' attention levels, measured by employing qEEG, showed that participants exhibited better-sustained attention while in VSW than in SBST mode. In addition, initial studies of the virtual training platform, designed to be adaptive to the user, are performed using deep learning and qEEG.

**Keywords:** online training; virtual world; slide-based training; fire safety; knowledge retention; engagement; attention qEEG; deep learning

## 1. Introduction

In the recent pandemic (COVID-19) situation, online training is preferred to allow learning with social distancing. Online training is gaining popularity for its advantages, such as portability, flexibility, and convenience [1]. Slide-based online training mainly consists of slides presenting information with sound, images, and/or video features. There are two types of slide-based online training: (1) instructor-facilitated and (2) self-paced. Many organizations have transitioned from hands-on and in-person training to computer-based training. It is less expensive than traditional training methods, and there is no requirement of hiring professional training staff [2]. In addition, it is possible to reach a large audience through such online training [3]. However, some disadvantages include little to no interaction between the trainee and slide-based instructions, making it a unidirectional flow of information [4]. Additionally, in the present era, the workforce is culturally diverse, young, and spread worldwide. Thus, training needs to accommodate multiple locations, learning styles, cultural diversity, and multiple languages [5].

Virtual reality (VR) is another form of online training that utilizes computers, software, and hardware to generate a simulated environment. It can engage a user by creating a sense of being present in that environment [6]. We define a virtual environment as an environment displayed on personal or desktop computers, while virtual reality applications that involve the use of sophisticated instruments, such as Head Mounted Displays, are not

in the scope of this discussion. Virtual world and/or game-based training is considered an environment for practice, interaction, and immediate application of knowledge by forming a mental model for newly learned information [7]. A virtual training environment allows dynamic decision-making, giving learners a better understanding of applying knowledge to real-world situations [8]. This process provides real-time feedback on the learner's specific choice and modifies the environment as per the learner's choices. Virtual simulation allows trainees to make mistakes and correct them in various scenarios that are difficult to recreate in the real world due to constraints of ethics, cost, and time [9]. People interacting with objects in a virtual environment develop an ability to interpret various relations, develop a habit of awareness and even think about surroundings before acting in real-world situations [10]. The virtual environment provides a combination of immersion and interaction in an environment, which mimics the real world [11]. Further, it allows participants to log interactions, provide feedback to trainees, and perform a behavior analysis [12]. However, high initial development costs and time are some of the disadvantages of virtual reality-based training.

In the COVID-19 pandemic situation, traditional education practices are switched to online mode. However, in some cases, such as fire safety and emergency evacuation training, this transition is posed with challenges. Traditionally, students are trained on health and safety in classroom settings, hands-on activities, and fire drills. From the view of such training as fire safety and emergency evacuation, it is challenging to make trainees visualize and practice finding exits and escape routes in slide-based training [13]. Though the fire drills' importance is undeniable, they are expensive to conduct multiple times with limited participants. Moreover, they may not present all aspects of an emergency or hazards to all participants [14]. Introducing an interactive health and safety platform that emphasizes self-learning can improve safety awareness and effectively influence behavior in real-life situations requiring the application of knowledge.

The objective of this study is (1) to develop interactive virtual training and (2) assess the effectiveness, attention, and engagement of this training against traditional training approaches, such as slide-based training. There have been incidences of fires reported in and around universities. From the year 2000 to 2015, 118 fatalities were reported from 85 fires that started in dormitories, fraternities, sororities, and off-campus housing in the U.S.A. [15]. We have selected fire safety and emergency evacuation training for the study to apply to universities and various industries. The contribution of this research over the previous studies is the assessment of factors such as effectiveness, engagement, and attention level, using a large sample size, utilization of qEEG to validate attention levels during conventional and virtual safety training and the initial exploration of deep learning [16] to create an adaptive interface to the trainee. Our results validated that a Virtual Safety World (VSW) can provide the same knowledge as slide-based safety training (SBST) but can do so in a more engaging manner. The VSW group voluntarily spent more time in training than the SBST group. Retention of concepts after a month was higher in VSW participants. The participants' attention levels, measured by employing EEG, showed that participants exhibited better-sustained attention while in VSW than in SBST mode.

The paper is structured as a literature review in Section 2, followed by the development of online training modules in Section 3. Section 4 describes the method used to conduct this study, including recording an electroencephalogram (EEG) of participants (n = 40). An empirical evaluation of this virtual environment compared to slide-based online training on effectiveness, engagement, and attention level is explained in Section 5. Section 6 describes a continuation of the experimentation for the development of an adaptive user interface using deep learning (DL). Finally, the paper concludes with a discussion and insights obtained from the study.

## 2. Literature Review

### 2.1. Virtual Environments in Education

In recent years, virtual simulation in education has become a topic of interest [17]. Virtual environments have been developed using web-based, multi-user platforms, such as Unity, Second Life, and Open Simulator. August et al. [18] developed STEM (Science, Technology, Engineering, and Mathematics) education activities in Second Life, while the OpenSim-based game is introduced in tertiary education as a supplementary training tool by Terzidou et al. [19]. In La Sabana's university, a case study showed that the Second Life-based virtual world could support in learning subjects such as electronics. It was observed that participants felt more involved and absorbed in the virtual platform, making the learning process effective. However, the platform was also responsible for causing distractions, as Second Life allowed students to access other websites with different contents and social networking [20].

### 2.2. Virtual Training and Fire Safety

Researchers have been developing virtual environments using different virtual training platforms for various industries to learn safety concepts. For example, Sacks et al. [9] built an environment for understanding the effectiveness of implementing virtual reality-based safety training in the construction industry. In their study, 66 subjects were divided into two groups. Half of the subjects received traditional, lecture-based training, and the remaining subjects were trained using a 3D, immersive, virtual-reality power wall [6]. Safety knowledge of the participants was tested before the training, immediately afterward, and one month later. Virtual training was more effective in maintaining the participants' engagement and in the long-term recall of knowledge. For example, Wener et al. [21] compared a web-based interactive game with traditional classroom firefighting training. Authors reported that the firefighters trained in a web game performed better on the post and long-term retention tests than those trained in a classroom.

Padgett et al. [22] used a virtual reality computer game to teach fire safety skills to children diagnosed with fetal alcohol syndrome. This virtual reality development was a good study, but the study group was reduced to five children. In addition, a complete longitudinal study was restricted to only one week. Smith and Ericson [23] trained children on a virtual simulator for reinforcing video-based fire safety training. Pre-quiz and post-quiz were utilized as an instrument to understand short-term learning gains in this study. The pre-and post-quiz analysis showed that short-term learning was not impacted positively or negatively due to training. Their study used 22 participants (a group that may be considered small). Their research did not consider longitudinal studies. Silva et al. [24] developed a preliminary 3D fire-evacuation simulation game. A sample of 20 healthcare professionals was selected to test the hypothesis of applying simulation-based training as an aid to a traditional fire drill. The preliminary results demonstrated the viability of the simulation approach. However, further research and development were found to be required to improve scenarios and playability, and for adding multi-player capability in the game.

Christoph et al. [25] described how a simulation environment is implemented for safety and security training in the maritime merchant field. Officers, crew, and service personal were offered simulation training that incorporated emergency scenarios, such as fire, flooding, and bridge evacuation. The training results showed that simulation-based training optimized emergency management training and improved team performance along with collaborative learning.

A 3D simulation game for fire safety skills was developed from the first-person view, where three floors of a university building, each consisting of 100 rooms, were designed by Chittaro and Ranon [14]. The authors reported a preliminary user test with seven subjects, and measurement of effectiveness, engagement, and transfer was identified as future work. Tawadrous et al. [24] developed a platform to train large institutions such as universities on threats such as toxic fire in the laboratory. A well-known game platform called Unity

was used to develop a 3D kitchen fire safety awareness game, which trains participant on using a fire extinguisher and how to recognize if they should call the fire department or evacuate their home during a fire emergency [26]. An OpenSim and Second Life-based virtual environment was developed [27] to train firemen trainees.

To summarize, the literature echoes that a virtual environment can act as a complementary tool in the learning process [18–20]. Further, researchers have tested it in fields, such as construction, firefighting, and healthcare for safety training. Researchers have also developed a university building-based 3D game for fire safety training, but a comprehensive evaluation of training has not been conducted [14,26–28]. Cicek et al. [29] surveyed high school students to understand their perception of VR in education. They stated that using a virtual environment is a new technology, and there is a need to analyze its use in education sufficiently.

## 3. Development of Online Training Modules

This section elaborates on the development of online training modules and their evaluation.

### 3.1. Interviews of Experts

We adopted a Lean Startup [30] methodology to obtain experts' unbiased opinions on the fire safety training system. The Lean Startup methodology advocates developing products/services based on validated learning, i.e., learning requirements from customers, asking for their feedback quickly, and improving the system under development. Major stakeholders—students, lab assistants, faculty and staff, Environmental Health and Safety (EHS) experts, and professionals from government and private organizations were interviewed. The interview process helped to understand the frequency of fire safety training, the audience of the training, and the level of knowledge expected by the trainees. In addition, 45 personnel of varied ages, races, and genders were interviewed. These interviewees were classified into three groups—students, professionals, and EHS experts. The feedback obtained from interviewees is summarized in Figure 1.

**EHS experts**
- Careless behavior is a major cause behind accidents, which can be changed with better retention of safety training.
- Virtual simulations can aid in training but it cannot replace existing training
- Information should be provided to humans in such a manner that they can form a mental model which could help in decision making.
- People are being trained with slides for long time. Introducing 3D virtual training could be interesting but challenging in terms of conveying concepts.

**Professionals**
- Completing same fire safety training every year in computer instructions form or lecture form is tedious.
- Repetitive and verbose nature of training makes it boring.
- Overtraining can make people neglect/ forget important safety related information specific to their job.
- Transfer of training is the biggest training challenge!
- Flexible timings for training are critical for busy professionals, merely making training mandatory doesn't ensure that employees pay more attention to it.

**Students**
- Not all students receive fire safety and emergency evacuation training.
- Students living in dorms participate once a semester in fire drill if they are present in the dorm at the time of drill.
- Research assistants in research lab receive mandatory fire safety training.

**Figure 1.** Learnings from the Lean Startup interview process.

During the interview process, it was observed that professionals and experts considered the use of a virtual environment for fire safety training beneficial. However, it was identified that there is a need to scientifically prove that virtual fire safety training and engagement can convey the same concepts that are equally easy to understand and retain as conventional SBST. The information obtained from students, professionals, and EHS experts was crucial in developing SBST and 3D virtual training for this study. Most inter-

viewees stated that universities and organizations utilize SBST with images made available on the web, and it logs the information of users.

*3.2. Virtual Safety World Module*

Unity 3D, Open Simulator, and Second Life are platforms used to develop virtual safety training environments [18–20,24,26,27]. The criteria for developing VSW are the simultaneous presence of players, customization of the player avatar, user data confidentiality, flexibility for frequent modifications in the model, low development cost, and time. Our approach for developing the VSW module involves the following: (1) developing a 3D model based on an actual floor plan of a university building and making improvements as per the feedback of experts, (2) developing tasks for fire safety and modifying them based on the feedback of experts, and (3) conducting a small user study for controlling avatars and modifying avatars according to their feedback. Unity 3D is a popular cross-platform engine for developing interactive games. In Unity 3D, the graphics quality is better. The model needs to be developed in different software, then imported in Unity and edited for scenario development as mentioned by Silva et al. [24]. This process may not be ideal for development involving frequent modifications in the model. Open Simulator [31] and Second Life allow building model and development scenarios and avatars in the same software platform. However, Second Life was observed to be prone to causing distractions due to social networking access and it lacks the component of content ownership, confidentiality, and data security [20,32]. Hence, Open Simulator was chosen to develop a virtual 3D model of an engineering building and simulate fire safety and emergency evacuation training scenarios.

The actual floor plan of the university building was imported to OpenSim as a baseline, using basic building blocks. A four-storied engineering building was developed. A complete 3D model of the college building consists of 6400 primitive blocks. Figure 2b shows the exterior view of this model. Textures were applied to walls, floors, doors, and wallpapers. Carpets were placed in the model at appropriate locations in accordance with the actual building.

The next step was to develop a scenario on fire safety and emergency evacuation based on expert interviews, the literature, and the university's evacuation procedure and safety training. Information about the emergency exits, finding the nearest exit, reporting an emergency, and avoiding use of the elevator during a fire emergency are covered in both training modules. An avatar is a 3D representation of a person in a virtual world (Figure 2a). The avatar is required for interacting with the objects in the virtual world. One can also control his/her avatar's appearance by choosing body parts, skin, hair, clothes, footwear, and accessories. Four intelligent agents are placed in the environment to help participants with navigation and completion of tasks. For example, an intelligent agent points the player toward the correct exit. Participants cannot distinguish between an intelligent agent and an avatar of another participant. The behavior of intelligent avatars is programmed using the Linden Scripting Language (LSL) in OpenSim. In VSW, feedback is provided after every task. When participants touched objects in virtual training, they received tasks, and their responses to these tasks were saved in an external database [33]. A dialog box opens on the screen, displaying a question and options from which a correct answer needs to be selected (Figure 2c). For example, an avatar can see a computer catching fire in VSW and is asked about the action that he or she would take in this case. Depending on the participant's response, the correct response in such a case is displayed on the screen.

The VSW was evaluated by a professional virtual world expert and two EHS experts. Their feedback was incorporated to improve the VSW. The textures of the carpet and stair mat, the wall color, and the movement speed of intelligent agents were modified. Dimensions of doors and posters were changed to be proportionate to walls, which can otherwise cause simulation sickness. A few college students were asked to control an avatar, choose clothes and accessories for an avatar, and navigate the virtual safety world. It was ensured that the interaction data were appropriately recorded in a database. According

to the feedback of students and experts, help was required in navigating or finding tasks and exits in some locations. Intelligent agents were placed in such locations. After making these modifications, the virtual world was made available to participants for this study.

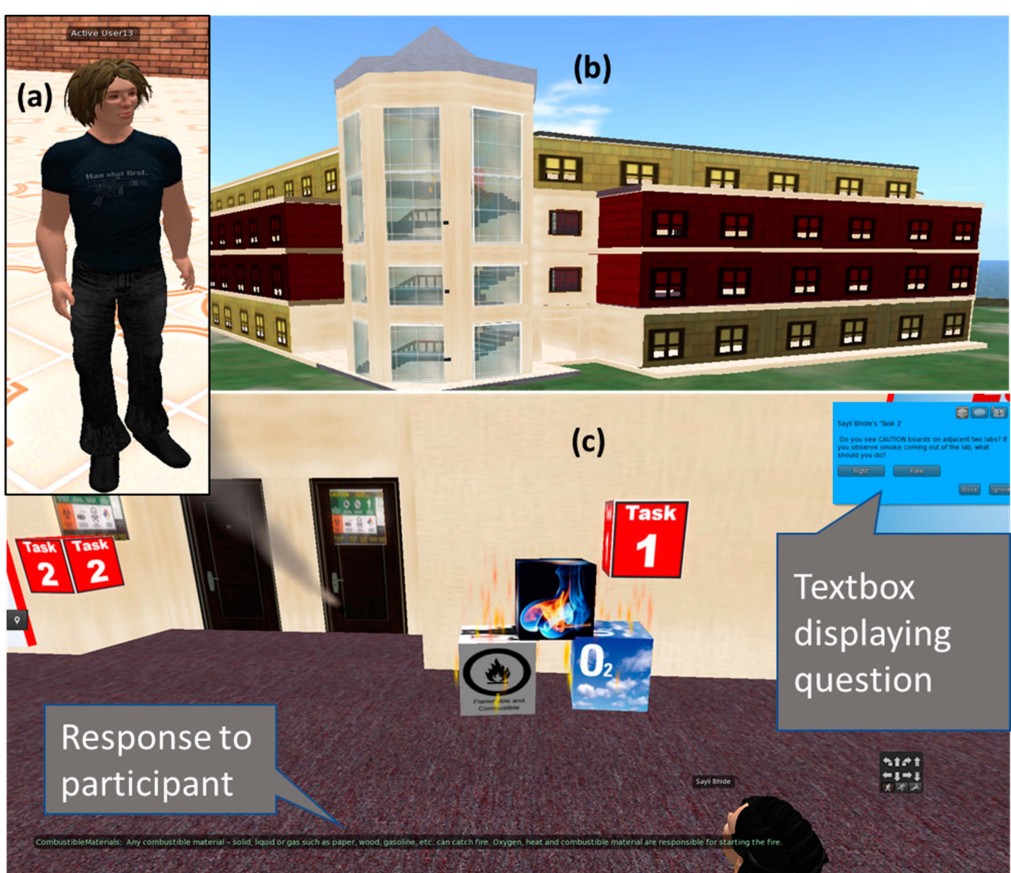

**Figure 2.** Details of the avatar and the virtual building built using OpenSim: (**a**) avatar of a participant; (**b**) virtual engineering building; (**c**) feedback in the VSW.

### 3.3. Slide-Based Safety Training Module

Slide-based safety training (SBST) is used in organizations where employees or students watch information in slides with sound, images, and/or video capabilities. This training is portable, as the trainee can watch information in any location and learn at his/her pace. From interviews of industry experts and professionals, slide-based training is a popular mode for training employees on health and safety in most organizations. Even in universities, the faculty, staff, and research assistants must complete computer instruction-based training on safety. Therefore, web-based training, consisting of slides, images, and data analytics, was chosen as a conventional training mode for the proposed study. Hence, it is termed slide-based safety training (SBST).

SBST was developed and made available on a website to trainees. The responses of the trainees to the questionnaire and the time spent by each trainee on each slide were saved. The slides developed consisted of appropriate images and five to six bullet points per slide. At the start of the training, a few introductory slides were included to explain the training and navigation objectives through the slides. Then, trainees were presented with a demographic questionnaire and pre-test followed by 32 slides on fire safety and the emergency evacuation procedure. Finally, trainees took a post-test and engagement questionnaire after the training session.

Two experts reviewed the SBST training developed. Slides were edited to accommodate changes as per the reviewer's feedback and to make it interesting for participants. For example, similar information on how to leave a building in case of emergency was

presented on two slides. One of the slides was removed after this was pointed out by one of the reviewers.

### 3.4. Evaluation of Training Modules

Training evaluation helps in understanding factors in the training that make it appropriate for the audience. Training evaluation is a continual and systematic process of assessing a training program's value or potential value. The results of the evaluation provide inputs to alter, continue, or eliminate the training components, such as design or delivery. The evaluation answers what occurred because of training [34]. The methods of assessment of training pertinent to this empirical evaluation are described below.

#### 3.4.1. Effectiveness

A knowledge test was developed to measure participant's' understanding of fire safety and evacuation procedures. It was based on Bloom's taxonomy, which consists of six major categories: knowledge, comprehension, application, analysis, synthesis, and evaluation [35]. The knowledge test had 12 questions mainly based on knowledge, comprehension, application, and analysis categories (refer to Appendix A). The assessment quiz on fire safety training, available on different university and industry websites, and experts' inputs were used to develop these questions.

The knowledge test was administered at three time points. At the beginning of the study, a pre-test was administered to capture the participants' knowledge baseline. The same test with a change of sequence in answer options and/or questions was provided as a post-training and final test. Participants could leave a question unanswered in case they did not know the concept. A correct answer to each question was credited 1 point, while a wrong or blank answer was credited 0 points. Hence, the maximum score possible on the pre, post, and final knowledge tests was 12.

#### 3.4.2. Engagement

An engagement questionnaire was provided at the end of the training, based on the questionnaires used in studies, such as [6,36]. Engagement-related questions focused on experience and concentration in the game. Trainees were asked to rate questions on a 5-point Likert scale (1 being strongly disagree and 5 being strongly agree). The time spent by the trainees in both training modules was recorded to understand their level of engagement. Cronbach's alpha value for the questionnaire was 0.701, which indicates an acceptable level of internal consistency. Two safety experts reviewed these instruments before they were used in the study.

#### 3.4.3. Attention Level

Attention is considered an essential characteristic for learning [37]. An electroencephalogram (EEG) is used in medical practice by physicians to observe patients' neural waveforms to detect an underlying abnormality. qEEG varies from conventional EEG as it performs signal analysis in the frequency domain, unlike time series [38]. Neural data collected in the time domain are converted into a frequency domain and classified into different frequency bands (delta (2–4 Hz), theta (4–7 Hz), alpha (8–12 Hz), and beta (13–30 Hz)) for analysis. qEEG is widely used in training research, for example, in the training of athletes and in yoga training to improve academic performance [39,40]. Figure 4b shows the EEG headset channel mapping concerning the brain. The beta band is considered to reflect the cortex's activation and is associated with increased concentration and task-related engagement levels. An increase in beta represents active processing [41]. When a person experiences higher engagement in a task, attention, focus, and concentration are increased. The ratio of power in beta to alpha (B/A) is observed to increase during brain activation [42] and used in this study to assess trainees' attention in both training pieces. The ratio of power in theta to beta (T/B) frequency bands is considered an indicator of

attention levels. High T/B was associated with low academic performance in medical college first-year students by Gorantla et al. (2018).

## 4. Method

Future research directions may also be highlighted. This study's protocols were approved in advance by the university's Institutional Review Board (IRB). There were 143 participants comprising students, faculty, and staff from the university. The majority of the study participants were undergraduate and graduate engineering students from departments such as industrial, mechanical and aerospace, computer science, electrical, and civil engineering. This selection allowed to gather a random sample representing the general population of students who usually spend time in the engineering college building. They chose the date and time slot as per their convenience for participating in the study. Date and time slots were randomly assigned to SBST and VSW to avoid a participant selection bias. After receiving the consent of the participants, they were presented with a demographic questionnaire and a pre-test.

The 143 participants were randomly divided into two groups. The first group of 73 participants was assigned to SBST on desktop computers, while the second group of 70 participants was trained on fire safety with a desktop-based, 3D virtual world. Participants were able to choose their avatar from 8 avatars (four male and four female). There was no time limit for completing training. After completion of training, a post test and questions about engagement experienced during training were administered to participants. Participants in both training groups were administered a pre-test, post-test, and engagement questionnaire (Figure 3). Participants from both training groups were asked to complete a final test (similar knowledge test as the pre- or post-knowledge test) online, after about a month from initial training.

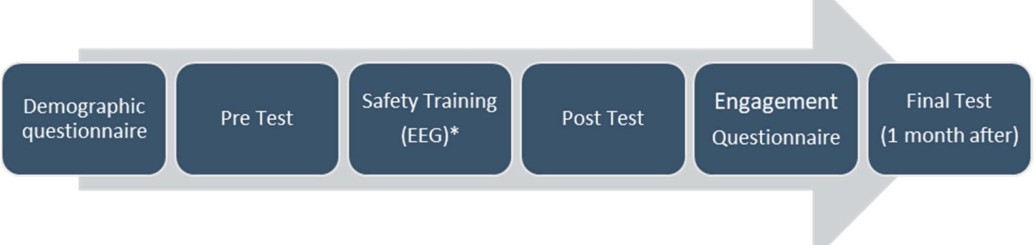

**Figure 3.** The procedure followed for SBST and VSW training (* 40 participants wore EEG headset while they were going through safety training).

The hardware for conducting the study consisted of 4 desktop computers (21 inLCD screen, NVIDIA GeForce GTX 770 (2 GB) VGA, Intel i7 processor, 16 GB RAM, mouse, and keyboard) for participants to undergo training. The standard OpenSim standalone region was rented, which provided 1024 MB of memory, an Intel quad-core processor to ensure high performance, and low lag in simultaneously displaying the virtual safety world. A laptop computer was used to record the EEG signal generated by the headset.

A 14-channel, non-invasive EEG headset was used to measure the neural signal powers of 40 participants (20 assigned to SBST and 20 assigned to VSW), randomly selected among the 143 participants. The EEG headset was placed on the participant's head, and their neural response was measured while undergoing fire-safety training (Figure 4a).

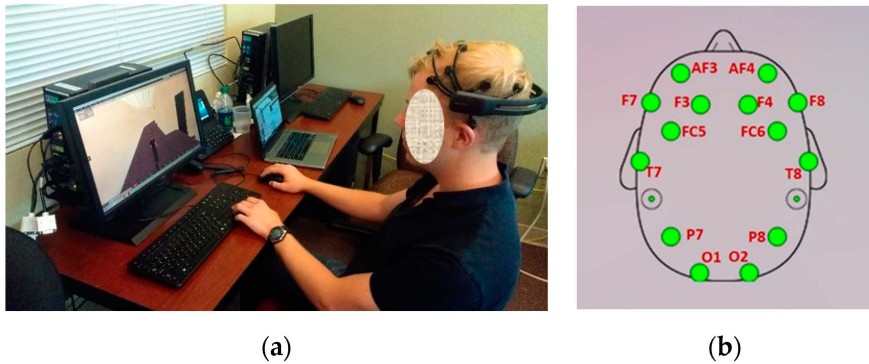

(**a**)                                                                                     (**b**)

**Figure 4.** EEG recording: (**a**) Participant undergoing VSW training wearing EEG headset; (**b**) channels mapping of 14 electrodes as per 10–20 international system.

- Procedure for data analysis of EEG.

The EEG signal recorded by electrodes from a person's scalp is a combination of neural signals and artifacts. Eye blinks, muscle movements, line noise, and the amplifier situation are significant sources of artifacts [43]. Thus, EEG data were first processed to remove artifacts from recorded data before performing an attention analysis. The procedure of data processing and analysis is explained below (refer to Figure 5).

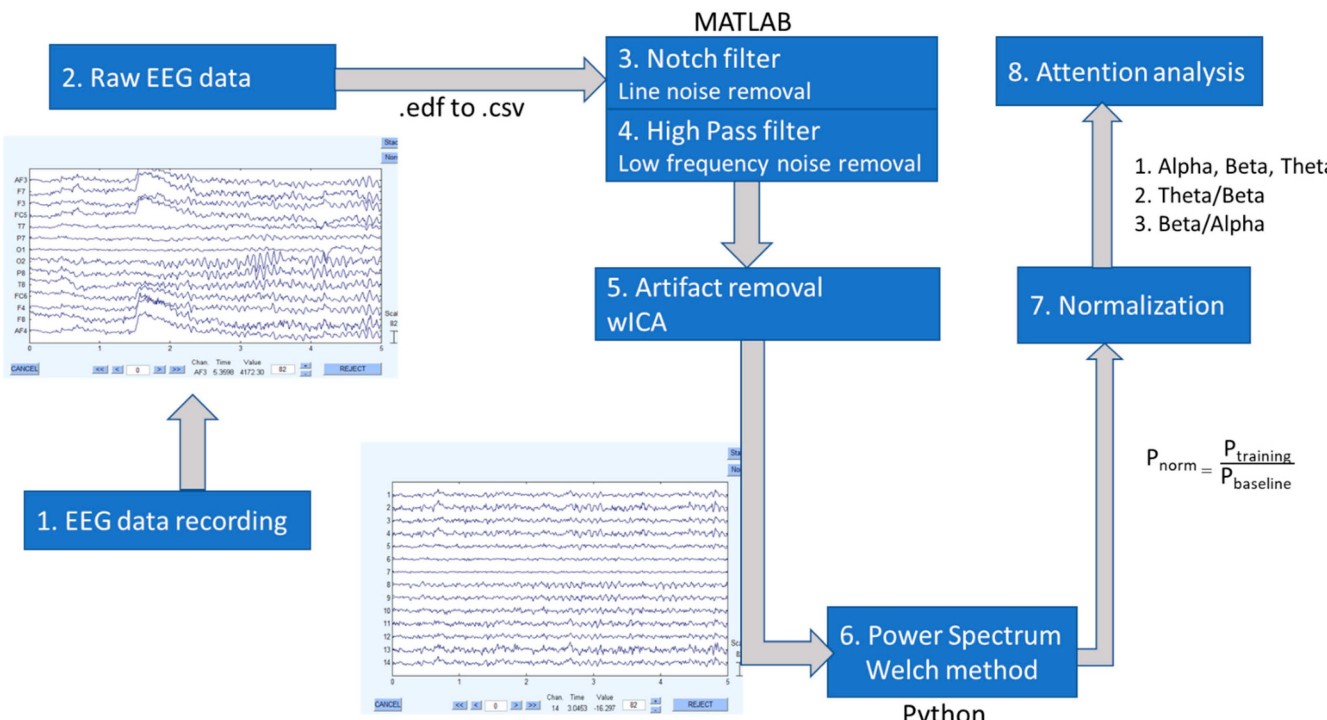

**Figure 5.** Data-processing steps for EEG recordings.

- EEG data were recorded in an EMOTIV testbench (1,2). MATLAB was used to remove line noise at 60 Hz by applying a notch filter, and a high pass filter was applied to remove low-frequency noise at 0.5 Hz from the recorded signal (3).
- The total amount of structured EEG data collected for 40 participants were analyzed using Python programming in steps 3 to 5. A wavelet-enhanced independent component analysis (wICA) method of artifact removal was used for denoising. All data in the .csv file from step (2) were divided into 15 s and wICA algorithms [44].

- The time-domain signal for each participant in each section was transformed into a frequency domain using Welch's method (6). This method employed a Hanning window, with segments of length 256 and 50% overlap.
- The raw power values are not normally distributed. Hence, the signal's normalization was achieved by using a baseline of eyes open signal (7).
- The power in each of the bands—delta band, theta band, alpha band, and beta band [45]—was calculated to perform the attention analysis. The mean power of the individual channel was used as an averaging technique for this analysis. In addition, the ratio of power in beta to alpha frequency bands was compared in SBST and VSW (8).

## 5. Results

### 5.1. Effectiveness

The comparison between SBST and VSW was statistically performed, using two-sample t-tests on the difference between knowledge test scores. (Please refer to Table 1 for an analysis summary.)

**Table 1.** Comparative effectiveness of training.

| Short Term Effectiveness | | | | | |
|---|---|---|---|---|---|
| Training Module | Sample Size | Pre Test Scores (Mean, SD) | Post Test Scores (Mean, SD) | ΔPost–Pre Test Scores (Mean, SD) | t-Stats, p-Value ($\alpha = 0.05$) |
| VSW | 68 | 7.28, 2.06 | 9.34, 1.68 | 2.05, 1.88 | $T = 0.26$ |
| SBST | 73 | 7.70, 1.93 | 9.53, 1.94 | 1.94, 1.80 | $p = 0.39$ |
| Long Term Effectiveness | | | | | |
| Training Module | Sample Size | Pre Test Scores (Mean, SD) | Final Test Scores (Mean, SD) | ΔFinal–Pre Test Scores (Mean, SD) | t-Stats, p-Value ($\alpha = 0.05$) |
| VSW | 46 | 7.11, 2.1 | 8.58, 1.91 | 1.48, 2.18 | $T = 2.19$ |
| SBST | 49 | 8.18, 1.7 | 8.85, 1.88 | 0.67, 1.76 | $p = 0.01$ |

- Short-term effectiveness: The difference between post and pre test scores (ΔPost–Pre) of participants for each training (SBST and VSW) was used as a metric to compare the short-term effectiveness of respective training. There was no significant difference in the ΔPost–Pre scores of a knowledge test for SBST and VSW ($p > 0.05$, $\alpha = 0.05$). Thus, the short-term effectiveness of both types of training was comparable.
- Long-term effectiveness: A total of 95 participants responded to the final knowledge test after about four weeks from the initial training. The difference between final and pre-test scores (ΔFinal–Pre) of the participants for each training method (SBST and VSW) was used as a metric to compare the long-term effectiveness. There was a statistically significant difference in the ΔFinal–Pre scores of a knowledge test for VSW and SBST ($p < 0.05$, $\alpha = 0.05$).

Statistical analysis of the knowledge test shows that individually, both SBST and VSW were effective immediately after training. However, a comparison between the one-month-after scores and pre-training scores of both types of training showed that participants from VSW remembered more after a month than participants from SBST.

### 5.2. Engagement

Seventy-two participants from the SBST group and 70 participants from the VSW group answered engagement experience-related questions. As the scoring was ordinal, a non-parametric test (Wilcoxon rank sum) was used to test the hypothesis.

The concepts presented in VSW were perceived as more engaging than SBST. VSW was a fun and enjoyable experience, compared to SBST. Participants felt that neither the slide-

based nor VSW environment had too much information, and it was not difficult to concentrate on concepts. Participants reported that they would like to undergo VSW next year, and it has modified their response to emergencies better than SBST (Table 2).

**Table 2.** Perception of participants on SBST vs. VSW.

| Question | Results ($\alpha = 0.05$) |
|---|---|
| The training material had significant, new content that you were not aware of. | $p = 0.0021$ |
| The training experience was fun and enjoyable. | $p = 0.0001$ |
| The training seemed to have too much information, and it failed to maintain your attention. | $p = 0.7667$ |
| It was difficult to concentrate on training material, and you felt distracted. | $p = 0.3818$ |
| You are likely to remember most of the key concepts presented in training a month from now. | $p = 0.0009$ |
| Taking this training has modified your likely response to a real-life fire/emergency evacuation situation. | $p = 0.0079$ |
| You would like to undergo the same fire safety and evacuation training next year. | $p = 0.0062$ |

Trainees in SBST and VSW could spend as much time as they wanted during training. There was no limit on how many times they could revisit slides/tasks during training. The data for time spent (in seconds) on training slides and tasks performed in the virtual environment were collected and analyzed. Two sample t-tests revealed that the time spent by trainees in completing tasks in the VSW environment (n = 68, Mean = 813 s, SD = 257 s) was significantly more than the time spent by trainees in SBST (n = 69, Mean = 580 s, SD = 315 s), ($p$-value = 0.001, $\alpha = 0.05$). Participants on both training modules were allowed to revisit the content. It was observed that in VSW, participants chose to perform certain tasks more than once. Due to the interactive mode of training, participants may have wanted to act on their feedback actions in VSW.

Statistical analysis of the engagement questionnaire showed that both training types presented information that was not considered difficult to concentrate on. This result supports that researchers did try to convey the same concepts in the best possible way for both training pieces. Additionally, evaluation of the training by subject matter experts helped to make both training types pertinent and exciting.

### 5.3. Attention Level

While the participant underwent training, the continuous neural signal was recorded. The signal was recorded in three phases: (1) for one and a half minutes of the participant closing their eyes; (2) next, for one and a half minutes of the participant looking at a blank screen with open eyes; and (3) while the participant was going through the training.

The mean power in alpha and beta bands for VSW participants in 'eyes closed,' 'eyes open' and 'training' modes is presented in Figure 6. In the relaxed state of mind with 'eyes closed,' power in the alpha band increased in participants. During the training phase, power in the alpha band reduced, while power in theta and beta bands increased during the 'eyes closed' phase.

The mean power, normalized with the 'eyes open' baseline in VSW participants and SBST participants, was plotted (Figure 7) for various EEG frequency bands. Normalized power in the beta band is observed to have significantly increased in VSW participants.

The ratio of the mean normalized power in the beta band to the alpha band (B/A) and theta to beta (T/B) were calculated for the entire duration of the training (Figure 8). The B/A of participants from SBST and VSW were compared statistically, using the two-sample t-test. Results showed that the B/A was higher for VSW participants than SBST

participants ($t = 1.96$, *p*-value = 0.02). The T/B in SBST participants was statistically higher than VSW participants on a two-sample t-test ($t = -3.87$, *p*-value < 0.0001).

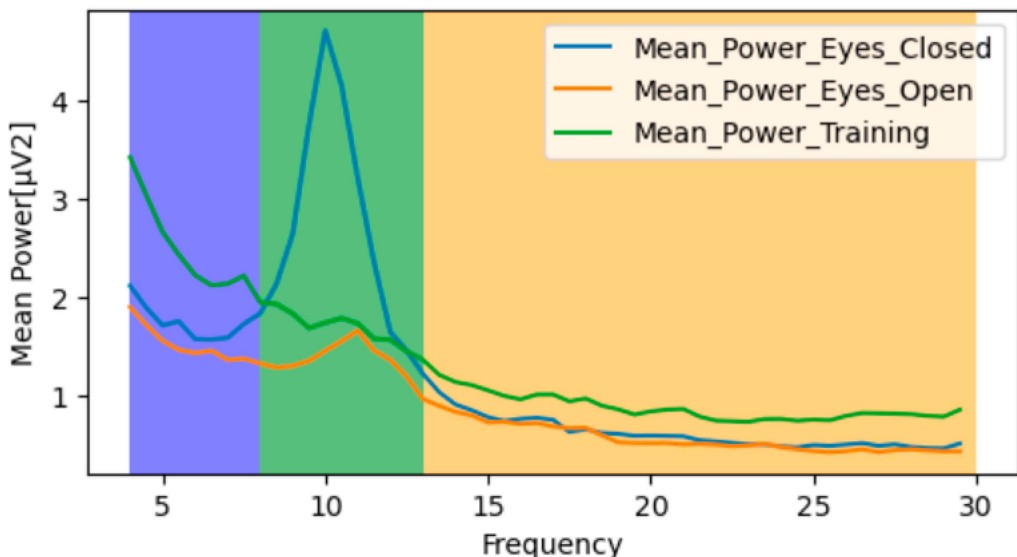

**Figure 6.** Illustration of mean power of participants in eyes closed, eyes open, and VSW training phases. Violet color represents theta, green color represents alpha, and yellow color represents beta frequency bands.

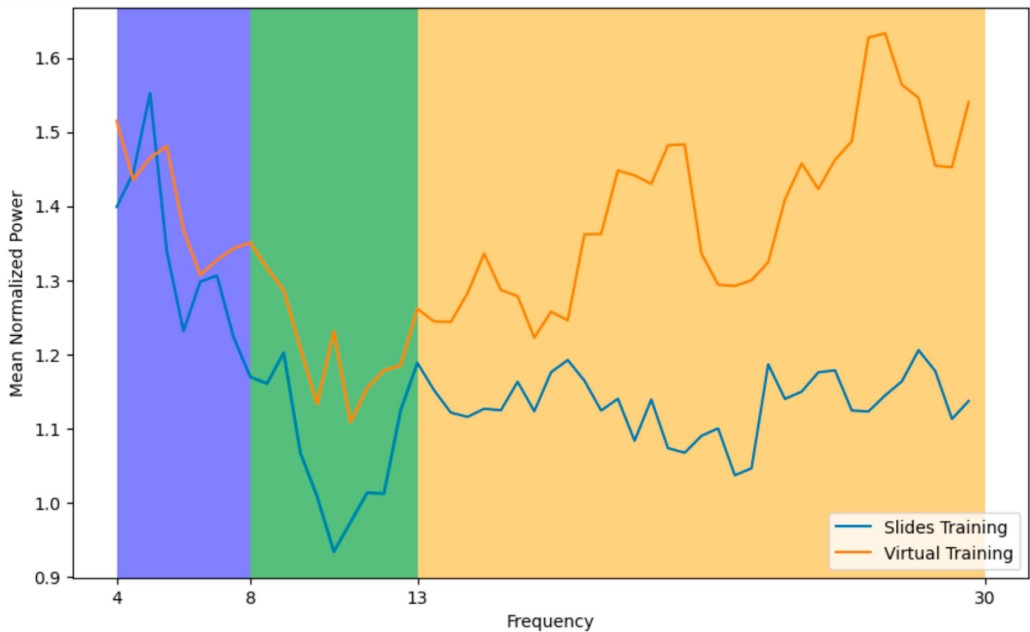

**Figure 7.** Mean normalized power in SBST and VSW on theta (violet), beta (green), and alpha (yellow) frequency bands.

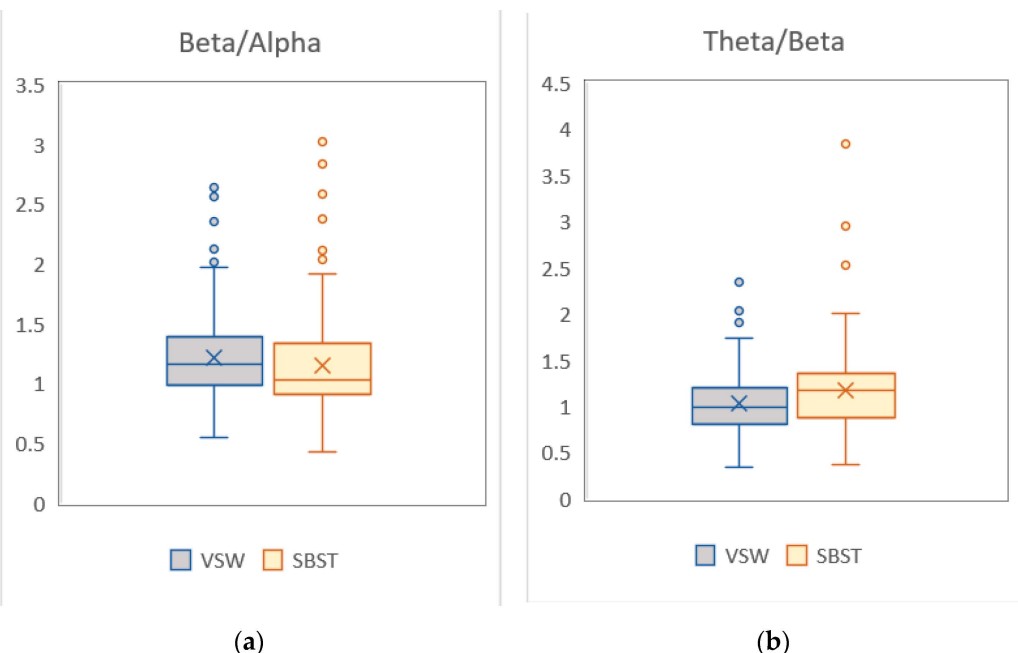

**Figure 8.** Attention level: (**a**) Beta/alpha ratio on VSW and SBST; (**b**) theta/beta ratio on VSW and SBST. Box plots represent distribution of the normalized power of VSW and SBST participants obtained from qEEG measurements during training. Symbol X represents mean.

## 6. Discussion

### 6.1. Effectiveness

Short-term knowledge retention was observed to be comparable in VSW and SBST training. However, long-term knowledge retention was better in VSW participants. Our finding on long-term retention of knowledge is consistent with the results of Wener et al. [21], reported in the study with firefighters. In addition, Sacks et al. [6] also reported that virtual reality-based safety training was more effective in long-term recall for construction workers than traditional classroom lectures.

### 6.2. Engagement

Participants in VSW perceived training to be more interesting than SBST participants. In addition, the perception of VSW participants about the long-term retention of training and transfer of training to real-life emergency evacuation situations was significantly better than SBST participants. This finding in our study is consistent with the study of [46]. The authors concluded that using a virtual environment to teach history increased interest and motivation and resulted in better long-term retention. In this study, one notable observation is that participants voluntarily spent more time and attempted tasks multiple times in VSW, compared to SBST.

### 6.3. Attention Level

Power in the beta band was observed significantly more in VSW participants than SBST participants. In addition, the B/A ratio was observed to be higher in VSW than SBST. This finding is consistent with Lim et al. [47]. The power in the beta band increases, the alpha band decreases, and the theta band increases during the state of attention and immersion resulting from playing computer games. The T/B ratio, measured by qEEG, is considered an indicator of attention control; high T/B indicates low attention. In this study, T/B was observed to be higher in SBST than VSW participants, indicating that participants' attention on VSW was better than SBST participants. This finding is consistent with the pilot study conducted by Gorantla et al. [38]. Authors reported that higher values

of T/B in participants were associated with lower attention levels, thus leading to lower academic performance.

## 7. Next Steps: Application of Deep Learning

We have taken advantage of the virtual world's development and the engineering analytics and EEG experience. We are now exploring the opportunities to modify the virtual world according to each user and support better learning. In this section, we report deep learning (DL) utilization to predict participants' verbal–visual ability.

Mayer and Massa [48] have stated that some people are better at processing words, while others are better at processing pictures. The recognition of a person's particular cognitive style is particularly relevant to the "design of multimedia training because multimedia training involves presenting words and pictures to learners" [48]. This specification is essential to support a better learning approach. Therefore, we would like to present the guidelines that better use the recognition of the cognitive style based on the EEG and some characteristics of the person, such as gender and age, and education level. We used the same virtual world, but there were modifications to the boards explaining the experiment's guidelines to the user. One of the guidelines has more prompts with graphics and pictures, while in the other one, text is more prevalent as presented in Figure 9.

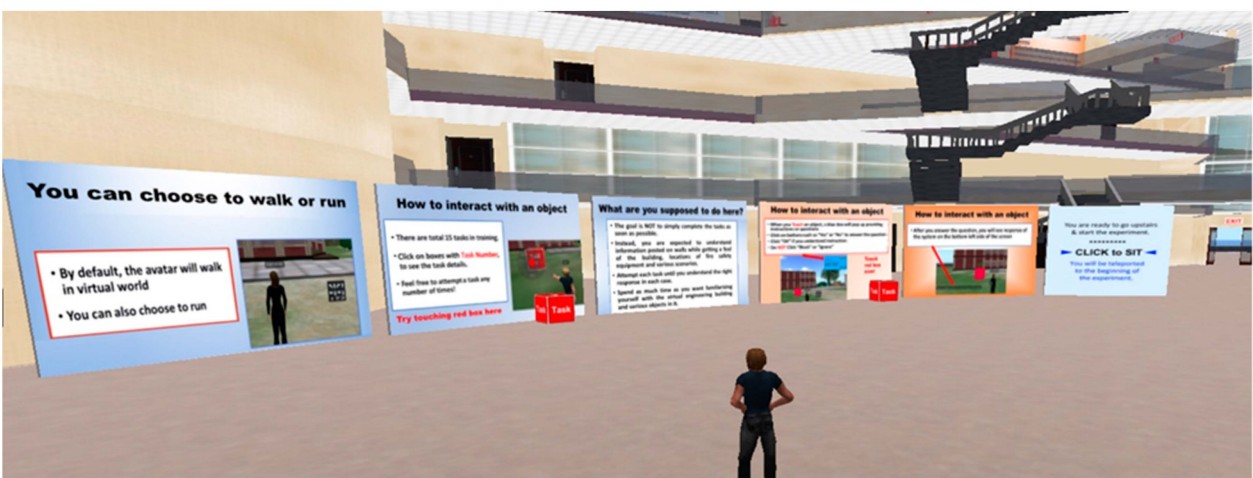

**Figure 9.** Emphasis on text to provide guidelines for the training.

Forty-six new participants were utilized: 23 for the environment emphasizing text information and 23 for the virtual world with more pictures in the guidelines. Information with the EEG was recorded during the session and divided into segments of 1 s. A participant was classified as a verbalizer, neutral, or visualizer before entering the virtual world [48–50]. This classification was matched to the respective EEG and demographic information (age, educational level, gender).

We utilized DL to create a mechanism based on EEG information to detect this trainee's cognitive style. DL, a subset of Machine-Learning techniques, was used for data analysis. DL algorithms allow computers to learn from examples. They have been successfully utilized in different domains to solve various problems. DL employs complex models to decrease errors in regression problems and increase classification accuracy. DL models' considerable learning capacity grants them the ability to make predictions and classification that are particularly useful [51]. Most DL methods utilize neural network architectures, so DL models are commonly called deep neural networks. "Deep" refers to the number of hidden layers in the system.

Traditional neural networks include 1 to 2 hidden layers, while deep neural networks include up to 150 hidden layers. Deep learning has several architecture/neurodynamic variations [52]. The variations are unsupervised, pre-trained networks (UPNs), convolu-

tional neural networks (CNNs), recurrent neural networks (RNNs), and recursive neural networks [53].

For this task, we decided to use a CNN. A deep learning architecture, using three dense layers, was applied to predict the participants' verbal–visual abilities. CNN is a multi-layer neural network, used to process data in multiple arrays [16]. The data were divided into three sets: one to train the network, another one to validate, and the final one to test the CNN developed.

The CNN network built can predict whether a participant is a verbalizer, neutral, or visualizer, with an accuracy of 83%. The results are encouraging despite the limitation of the data. Further development of this approach can provide an effective mechanism to modify the environment to support better learning dynamically.

## 8. Conclusions

In this study, two online training modes were developed on fire safety and emergency evacuation for the university community. Training material and an assessment questionnaire were developed with the help of university and industry experts. Further, two training types were compared on factors such as short-term and long-term knowledge retention and engagement experience. The results validated the literature findings that VSW provides an engaging, compelling, and interactive learning environment [6,18,20]. Virtual safety training provides the same knowledge as slide-based training and does so in a more engaging manner. Moreover, its long-term effectiveness was proven to be better than conventional SBST.

This research provides evidence that people can be trained on the same fire safety and emergency evacuation content more engagingly and effectively, using an interactive 3D virtual world compared to traditional method. Thus, this virtual 3D training can be utilized to train residents on the basics of fire safety and emergency evacuation without any state-of-the-art equipment in an engaging manner.

Virtual fire safety training platforms can help introduce college students to various skills and scenarios before joining the industry. Further, such a platform can be provided in organizations with different, industry-specific scenarios to learn about workplace hazards in a safe environment.

In this phase of the study, the focus was to train participants on basic fire safety and emergency evacuation in a virtual building of the university. VSW can be extended to other buildings and settings, such as hospitals, stadiums, and care homes, by considering respective safety protocols. Further study needs to be conducted to establish the efficacy of the virtual training in other settings. Crowd simulation can be thought of as the next step of this work. Multiple simultaneous participants and NPCs can participate in evacuation scenarios to understand the social aspects and interactions involved when crowds are evacuating under emotional distress.

This study's limitation is that though participants can visualize and practice fire safety concepts in a risk-free virtual environment, it does not shed light on how they react in a real emergency. Thus, it is challenging to measure the transfer of training. We plan to introduce multiple levels with increasing difficulty in virtual simulations to test understanding and to record participants' response times in finding correct exits. In future studies, other platforms, such as Unity3D, can be used in development to add more flexibility. The EEG study sample was small due to resource constraints, such as voluntary participation, experiment set-up time, and the one-on-one method of collecting neural signals. Though the sample size of participants in the EEG experiment was consistent with studies in the literature, there is a scope to conduct an attention study with a larger sample size in the future.

From the viewpoint of DL utilization, we will continue improving the cognitive style's prediction and dynamically changing the interface. We are also starting to detect simulation sickness and the level of attention online—this will be reported in future papers.

**Author Contributions:** Investigation, S.S. and F.A.; methodology, L.R.; validation, K.N. All authors have read and agreed to the published version of the manuscript.

**Funding:** This research received no external funding.

**Institutional Review Board Statement:** The study was conducted according to the guidelines of the Declaration of Helsinki, and approved by the Institutional Review Board (or Ethics Committee) of University of Central Florida (IRB number: SBE-16-12056 and date of approval: 02/22/2016).

**Informed Consent Statement:** Informed consent was obtained from all subjects involved in the study.

**Data Availability Statement:** The data presented in this study are available on request from the corresponding author.

**Acknowledgments:** Authors would like to thank experts from the industry and university for reviewing virtual fire safety and slide-based training and providing valuable feedback. Authors are thankful to the i-Corps team for training authors on the Lean Startup method.

**Conflicts of Interest:** The authors declare no conflict of interest.

## Appendix A

Questionnaire:

1. Which of the three elements are essential to start a fire?

    a. Oxygen
    b. Carbon dioxide
    c. Heat or source of energy
    d. Any combustible material

2. In case of a fire emergency in a building, you should

    a. Use an elevator to get out quickly
    b. Look for the nearest exit route
    c. If you are outside, quickly enter the building to grab your valuables
    d. Start reading instructions to use a fire extinguisher

3. You should never fight a fire if

    a. There is a risk of toxic fumes or explosion
    b. You have no idea what is burning
    c. You do not know which fire extinguisher to use
    d. The fire is too small and not spreading

4. Fire extinguisher generally lasts

    a. Few seconds
    b. Few minutes
    c. About an hour
    d. Few hours

5. An ABC-type fire extinguisher should be used against which of the following types of fires?

    a. A fire involving burning magnesium and plastic
    b. A fire generated when somebody poured water in a container that had sodium stored in kerosene
    c. A fire involving wood, gasoline, and electric saw

6. What should be your course of action in the following situation: you see fire coming out of the ceiling, there is a safe evacuation path behind you, and visibility is good.

    a. Flee
    b. Fight
    c. Call a friend for an opinion
    d. Wait for someone's instructions

7. You saw smoke coming out of electric wiring. You want to alert others in the building. What are the common locations of the fire pull station in the Engineering 2 building?

    a. Next to the elevator
    b. Next to the restroom
    c. Next to the water fountain
    d. Next to Exit doors

8. If you choose to fight a fire, where should you position yourself?

    a. As far away from the fire as possible to avoid getting hurt
    b. Next to a window so you can get out of your efforts to extinguish the fire are unsuccessful
    c. Six to eight feet from the fire, between the fire and your escape route
    d. As close to the fire as possible to ensure maximum efficiency of the extinguisher

    Read the following case and answer questions 20 to 23.

    When a student was using a microwave oven, it caught fire. She decided to fight the fire and left the room in search of a portable fire extinguisher. She returned after a few minutes, and in a rush, she tried to scan the instructions on the extinguisher and applied the PASS (Pull, Aim, Squeeze, Sweep) method. The fire seemed to diminish but did not extinguish.

9. What is the most likely reason the fire did not extinguish?

    a. The student did not turn off the power supply
    b. The student applied steps of the PASS method in the incorrect order.
    c. The student did NOT quickly pour water on the fire from a nearby tap
    d. It is not possible to extinguish fires involving electrical equipment with a portable fire extinguisher

10. Identify the CORRECT approach for extinguishing the fire in this case.

    a. Use Class B fire extinguisher
    b. Pour water from the nearby tap
    c. Use Class D fire extinguisher
    d. Use Class ABC fire extinguisher

11. Identify the CORRECT reason to fight the fire in this case.

    a. Since the fire started when the student was using a microwave oven, she felt obliged to extinguish it
    b. The student felt she could be a hero if she extinguishes the fire on her own
    c. The fire had just started and was small
    d. She did not want to bother everyone in the building by pulling the fire alarm

12. Instead of fighting fire if the student had decided to flee, what should be her very FIRST step?

    a. Exit the building and reach a safe place away from the building
    b. Leave the room and close the door behind her to confine the fire
    c. Activate fire alarm system to notify occupants about the fire
    d. Call 911 or university police by standing next to the fire

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
