# Peer review of "Virtual World as an Interactive Safety Training Platform"

_information, doi:10.3390/info12060219_

Round 1
Reviewer 1 Report
This paper investigates virtual worlds as safety learning platforms in a scenario of fire emergency evacuation of a university building.
The research develops two online training modes- slide-based and virtual world and assesses them on factors like knowledge retention, engagement, and attention. Empirical evaluation was conducted with 143 participants, with positive results.
The paper is well-written and the research is well-organized.
However, I have several questions and concerns about the novelty and the scope of the research goals and approach.
1. Why university? It is not clear why a university was selected for the training and simulation. Were there reasons of convenience for this decision? Or, is it that university evacuation presents important requirements and issues that must be dealt with in evacuation training? What are these requirements and how have these been considered in the training and evaluation scenario? Is the approach dependent on universities or can be applied to other contexts, like for example stadiums, children camps, hospitals, elderly care homes, ships, etc.?
2. Crown simulation? Universities are crowded buildings. Considering crowd behaviors in evacuation scenarios is an important factor of this research. Have you considered crowd behavior into the training and simulation scenarios? If so, how many people were in the crowd, and what type of behaviors were tested?
3. About scenario, context and content design. Is there an evacuation plan in this particular university? Was there an evacuation plan in the training? What were the main assumptions? E.g. Were there trained personnel available in the virtual word, to guide the people during emergency evacuation (as it would be expected, in some universities, but also in other places like hospitals or ships)? Were they the real personnel of the university in the virtual world, or perhaps NPCs? What about the current wayfinding design of the virtual university - how was that designed into the virtual model and was it considered for the evacuation scenario and tasks? Were the participants members of the university?
4. About novelty. Virtual worlds have been assessed positively for many training use cases in the past, including emergency fire evacuation.. What is different in this research, and what new contributions does it add to the current knowledge in the field?
5. What is the outlook of this research? What more needs to be done in order to see this virtual world / VR training of building evacuation, like stadiums, hospitals, universities, etc., incorporated in current safety training and related protocols?
Reviewer 2 Report
First of all, I would like to mention that I really enjoyed reading your manuscript.
In spite of the fact that I like it, I have some questions:
Page 6, lines 218- 223: "Four intelligent agents are placed in the environment to help participants in navigation and completion of tasks. For example, an intelligent agent points the player towards the correct Exit way. Participants cannot distinguish between an intelligent agent and an avatar of another participant. The behavior of intelligent avatars is programmed using the Linden Scripting Language (LSL) in OpenSim." is written. Question to this section: These sentences suggest that the user's avatar can interact with other participants in this virtual world. How many users can participate in this environment at the same time? Was it a limit or not?
Page 7, lines 272-273: "A knowledge test was developed to measure participant’s understanding of fire safety and evacuation procedures, a." is written. it seems that this sentence is not finished.
Page 11, line 439-440: "SBST participants did not revisit slides, but VSW participants chose to perform the tasks more than once." is written. Are you sure that it is not influenced that the VSW group remembers more?
Other questions:
How long did it take to develop the virtual environment?
How many days or weeks did SBST or VSW training last?
How long participants did learn all "lessons" or necessary information?
To sum it up, this manuscript is written well, I suggest it for publication.
Reviewer 3 Report
Il contribution presents important and original contents, with a presentation of solid scientific quality and the general merit is certainly high.
Il contribution provides an adequate background that includes references from relevant scientific literature.
The research presented is well described, appropriate for the journal, with clearly described and presented methods and results. The conclusions are also well supported by the results.
Author Response
Authors are thankful to reviewer for the review and the feedback.
Round 2
Reviewer 1 Report
The authors have addressed my comments well enough. I suggest acceptance of the paper.